# An Effective Lunar Crater Recognition Algorithm Based on Convolutional Neural Network

**Song Wang [1], Zizhu Fan [1],\*, Zhengming Li [2], Hong Zhang [1] and Chao Wei [1]**

[1] School of Science, East China Jiaotong University, Nanchang 330013, China; 2018088070100005@ecjtu.edu.cn (S.W.); 2018088070100004@ecjtu.edu.cn (H.Z.); 2017088070100004@ecjtu.edu.cn (C.W.)

[2] Industrial Training Center, Guangdong Polytechnic Normal University, Guangzhou 510665, China; gslzm@gpnu.edu.cn

\* Correspondence: zzfan@ecjtu.edu.cn; Tel.: +86-139-7099-3650

**Abstract:** The lunar crater recognition plays a key role in lunar exploration. Traditional crater recognition methods are mainly based on the human observation that is usually combined with classical machine learning methods. These methods have some drawbacks, such as lacking the objective criterion. Moreover, they can hardly achieve desirable recognition results in small or overlapping craters. To address these problems, we propose a new convolutional neural network termed effective residual U-Net (ERU-Net) to recognize craters from lunar digital elevation model (DEM) images. ERU-Net first detects crater edges in lunar DEM data. Then, it uses template matching to compute the position and size of craters. ERU-Net is based on U-Net and uses the residual convolution block instead of the traditional convolution, which combines the advantages of U-Net and residual network. In ERU-Net, the size of the input image is the same as that of the output image. Since our network uses residual units, the training process of ERU-Net is simple, and the proposed model can be easily optimized. ERU-Net gets better recognition results when its network structure is deepened. The method targets at the rim of the crater, and it can recognize overlap craters. In theory, our proposed network can recognize all kinds of impact craters. In the lunar crater recognition, our model achieves high recall (83.59%) and precision (84.80%) on DEM. The recall of our method is higher than those of other deep learning methods. The experiment results show that it is feasible to exploit our network to recognize craters from the lunar DEM.

**Keywords:** crater recognition; DEM; convolutional neural network; effective residual U-Net

## 1. Introduction

The Moon is the closest celestial body to Earth and the natural satellite of Earth. There are a lot of impact craters on the surface of the Moon. Impact craters and highlands constitute the typical lunar landform. While the Moon lacks common terrestrial geological processes due to atmosphere, wind, and water, impact craters can be well preserved. The study of lunar landform and lunar rock structure via impact craters on the lunar surface is of great significance to the investigation of the origin and evolutionary history of the Moon [1]. The impact crater can be divided into two principal categories: the main crater and secondary crater. Secondary craters are small ones that are formed by the ejecta falling down the planet surface where big impact craters are generated. Hence, secondary craters tend to cluster in a discrete ray around the main crater. A large number of secondary craters can be found on the Moon, Mars, and other celestial bodies, where there is nearly no weathering. Secondary craters around the small main crater are usually so small that are hardly recognized, which is still a difficulty in the crater recognition. At present, the main method of crater recognition depends on

human observation. The advantage of the human observation method is that it can accurately classify various types of impact craters. However, the method of human observation is very time-consuming. It can be used to deal with image and video data. But human observation is not suitable for massive lunar exploration data, particularly massive lunar digital elevation model (DEM) images. Moreover, this method lacks a uniform objective criterion. Different people have different recognition results for the same lunar landform image. Even the same person may have different recognition results for the same lunar landform image at different times. The differences of experts reach on crater recognition of up to 45% [2]. How to quickly and accurately recognize lunar craters is still a challenging problem in the field of lunar exploration.

In recent years, in order to obtain more information on impact craters and reduce the consumption of human resources, a number of researchers have begun to design crater recognition algorithms. In 2005, Kim [3] proposed a method to automatically extract the crater features on Mars. The whole process consists of three stages: the focus stage, location stage, and debugging stage. This method can classify salient craters well. However, it has high computational complexity and cannot well recognize the craters with the complex distribution. Goran [4] proposed a new crater recognition algorithm based on fuzzy edge detectors and Hough transform on the DEM of Mars. In 2011, Ding [5] presented a universal and practical framework that used the boosting and transfer learning method to recognize sub-kilometer craters. Yan Wang [6] integrated the improved sparse kernel density estimator into the boosting algorithm and proposed a sparse boosting method. This method is used to automatically recognize sub-kilometer craters in high-resolution images. Although the sparse boosting method has low computational complexity, it can hardly classify the overlapping impact craters, and the obscure craters are easily regarded as the background. Besides, M.J. Galloway [7] proposed an automatic crater extraction method in 2015. This method uses the Hough transform, Canny edge detection algorithm, and the non-maximum suppression (NMS) algorithm to perform the crater recognition. Kang [8] proposed an algorithm to extract and identify crater on CCD stereo camera images and associated DEM data. The algorithm is based on 2-D and 3-D features, which extract geometric features from optical images, and finally, get the result by using 3-D features from DEM data. Vamshi [9] used an object-based method to detect craters from topographic data. The method uses high-resolution image segmentation to create objects and then shape and morphometric criteria to extract craters from objects. Zhou [10] extracted slope of aspect values at crater rims and applied the morphological method to get true crater shapes. In [11], Rodrigo Savage used the Bayesian method to analyze the shape of sub-kilometer craters in high-resolution images. This work has developed a parametric model using the diameter of the crater, the height of the crater edge, edge eccentricity, and direction, etc. to represent craters. Min Chen [12] detected lunar craters from lunar DEM based on topographic analysis and mathematical morphology. The algorithm can detect dispersal craters and connected craters, but it is not suitable for detecting overlapping craters because of the irregular distribution of the sink points of those craters. Due to the shortcomings of mathematical morphology, this algorithm performs worse on craters with incomplete edges. Chen Yang [13] used a deep and transfer learning method to identify and estimate the age of crater in Chang'E data and provided a two-stage craters detection method. In 2020, Lemelin [14] used the wavelet leaders method to get a near-global and local isotropic characterization of the lunar roughness from the SLDEM2015 digital elevation model. Almost all of the above methods have considered craters as a whole, which may fail to deal well with the incomplete and complex impact craters. In order to address this problem, we pay more attention to the edges of the impact craters, recognize them, and then get their location and size in this work.

Recently, deep learning has achieved great success in the field of computer vision. The algorithm based on the convolution neural network (CNN) is excellent in solving the problems of object detection, image segmentation, and image classification, etc. The CNN is a kind of feedforward multi-layer neural network with convolutional computation and deep structure. It is one of the representative algorithms of deep learning [15,16]. CNN reduces the data dimensions and extracts data features by convolution, pooling, and other operations step by step. It adjusts the convolution weight in the

training process. In 2015, Jonathan Long et al. [17] proposed the fully convolutional network (FCN) in which the full connection layer of CNN was replaced by the convolution layer. The output of FCN is the spatial domain mapping rather than the probability of categories. Thus, FCN transforms the image segmentation problem into an end-to-end image processing problem. Both the input and output of the network are images. It takes less than 0.2 s to segment a typical image. FCN has become a very important work in the field of semantic segmentation. After FCN, many models based on FCN (U-Net [18], R-FCN [19], SegNet [20]) have also achieved good results in image segmentation, object detection, and other applications. In [18], Olaf Ronneberger improved FCN and proposed a new semantic segmentation network called U-Net that was first applied to biomedical image segmentation. U-Net is an efficient neural network that can achieve good performance in neuronal structure segmentation. He et al. [21,22] proposed a deep residual network (ResNet), which sufficiently used the image information but did not add additional parameters. The problem of the degeneration in deep CNN is successfully solved by ResNet.

Lunar landform images can be roughly grouped into two categories. One category is the ordinary optical image. Another category is the DEM dataset. Since the DEM dataset is hardly affected by the illumination and camera angle, our work has performed crater recognition on DEM. Note that directly using image recognition methods mentioned above [5–7] cannot achieve desirable recognition results. In order to develop a crater recognition approach for the lunar DEM data, we exploit deep learning theory to propose a semantic segmentation network structure termed effective residual U-Net (ERU-Net) that borrows the ideas of U-Net and ResNet. It can detect the edge of craters well and perform better when the depth of network layers increases. ERU-Net uses U-Net as the basic network structure in which the traditional convolution is replaced by the residual convolution. In the proposed crater recognition algorithm, we firstly train an ERU-Net model to detect the edge of the craters. Then, we use a matching algorithm to get the crater location information from the edge detection result. Finally, we verify whether the extracted craters are real craters. Different from traditional machine learning methods, our method exploits the full convolution network to classify each pixel in the image. Even if the impact crater is incomplete in the lunar image, the edge of the impact crater can be accurately recognized by using the convolution network.

## 2. Materials and Methods

### 2.1. Experimental Data

We use the lunar digital elevation model (DEM) images marked by NASA [23]. The reason for using the DEM dataset is to avoid the influence of the illumination and viewing angle on the experimental results. The size of the image is 92160 × 30720, which spans 0 to 360°E and 60°S to 60°N. The DEM has a resolution of 256 pixels/degree or 118 m/pixel as shown in Appendix A. In theory, the full convolutional neural network can accept images of any size, but it is limited by the size of GPU graphics memory. Therefore, we clip the lunar DEM image to a number of images of size 256 × 256. Then, we use Cartopy [24], which is a Python package to transform the image into an orthographic projection, in order to minimize image distortion. It can avoid craters appear non-circular at high latitudes. After the training data are generated, the corresponding labeled data need to be generated. Because the impact craters are mostly circular, we draw circles in a 256 × 256 blank image as edge label of impact craters based on the lunar crater position information provided by Head [25] and Povilaitis [26]. Head provided dataset of lunar craters, which was larger than 20 km in diameter, by using high-resolution altimetric measurements of the Moon. Povilaitis provided a dataset of smaller craters, which was 5 to 20 km in diameter, by using CraterTools extension [27]. The total number of craters in Head's dataset is 5186, and in Povilaitis's dataset is 19,337. The validation dataset and test dataset are generated in the same way. The area of generating testing images is different from that of the training images, which ensures that the testing images are not within the training and validation dataset. From Figure 1, it can be found that the labeled dataset is not complete because the radius

of an impact crater in the dataset is greater than 5. For example, some small and shallow craters are missed, and even some of the obvious craters are not labeled. In general, increasing training data can obtain better recognition results. We generate 30,000 images as a training dataset and 3000 images as a validation dataset to train CNNs in experiments. Then, we evaluate the model performance on the test dataset containing 3000 images. The size of the image we crop is 500 to 6500 pixels. Then, we resize it to 256 × 256. The experiment data has a different resolution range from 230 m/pixel to 2996 m/pixel or 10 pixel/degree to 131 pixels/degree. As shown in Figure 1, the bottom right corner of each graph, the four images have different resolutions.

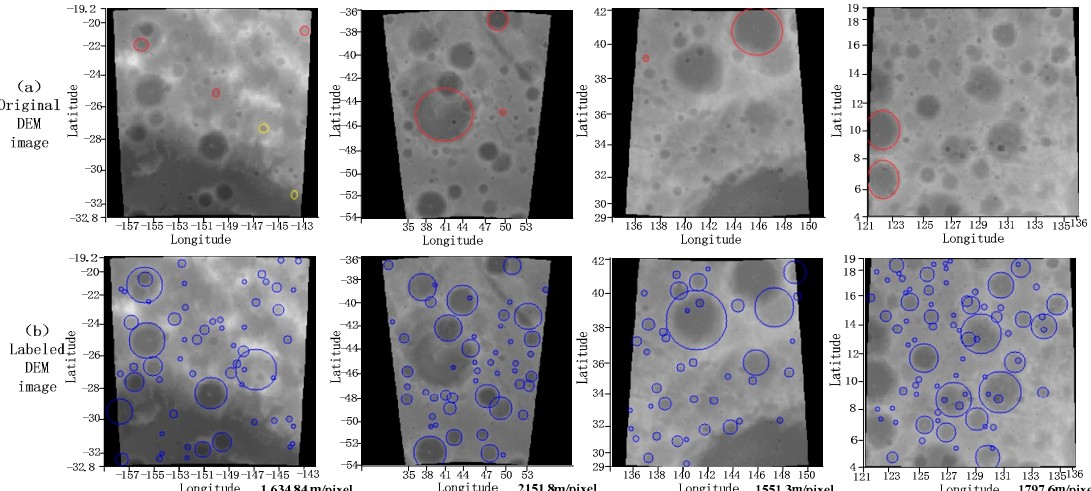

**Figure 1.** Original images and labeled images. (**a**) is the original image. (**b**) is the labeled image, yielding the ground-truth crater set here. The red circles in original images are the impact craters with obvious missing marks, and the yellow circles in original images are the impact craters with incorrect mark. The bottom right corner reflects the scale bars of the labeled DEM image. It means a pixel in the image corresponds to the real size.

## 2.2. Network Architecture

In this subsection, we design a new network called effective residual U-Net (ERU-Net), which uses U-Net [18] as network infrastructure, and combine the characteristics of residual blocks in ResNet [21], which can simplify network training and sufficiently use image information. The network structure of ERU-Net is shown in Figure 2.

As shown in Figure 2, the whole network is divided into three parts, i.e., the encoder, bridge, and decoder parts [28]. The encoder part encodes the input image into the tensor form, shrinks the size of the feature map, and increases the number of feature map channels. The bridge part connects the encoder and decoder parts. The decoder part changes the image features to pixel-level images, enlarges the size of the feature map, and reduces the number of feature map channels. The lunar landform images are input into the encoder part, and the last layer of the network outputs the lunar crater edge prediction results (see the first row in Figure 3). The encoder part contains three residual units [29], each of which is formed of a convolution layer and a residual convolution block. Each residual unit in the encoder is followed by a pooling layer to reduce the size of the feature map. There is a residual unit in the bridge part. The decoder part contains three residual units. Before each residual unit in the decoder, a deconvolution layer is added to enlarge the feature map. Then, the enlarged feature map obtained by each deconvolution is concatenated with the feature map of the same size and channels in the encoder part.

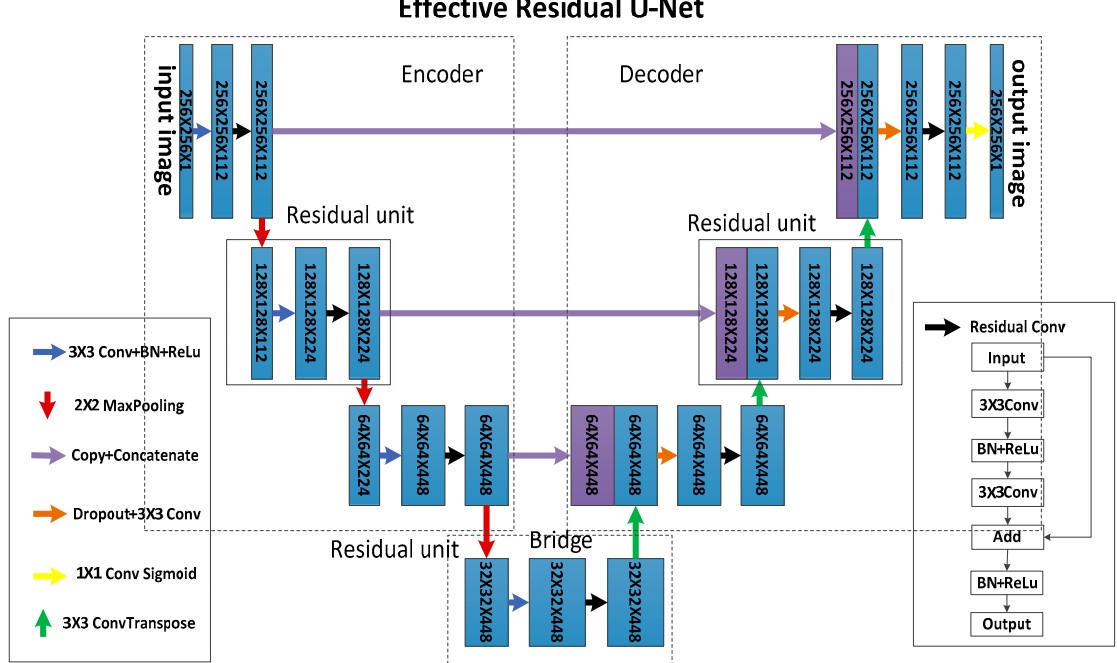

**Figure 2.** ERU-Net (effective residual U-Net) structure (the number of initial filters: 112, input image size: $256 \times 256 \times 1$).

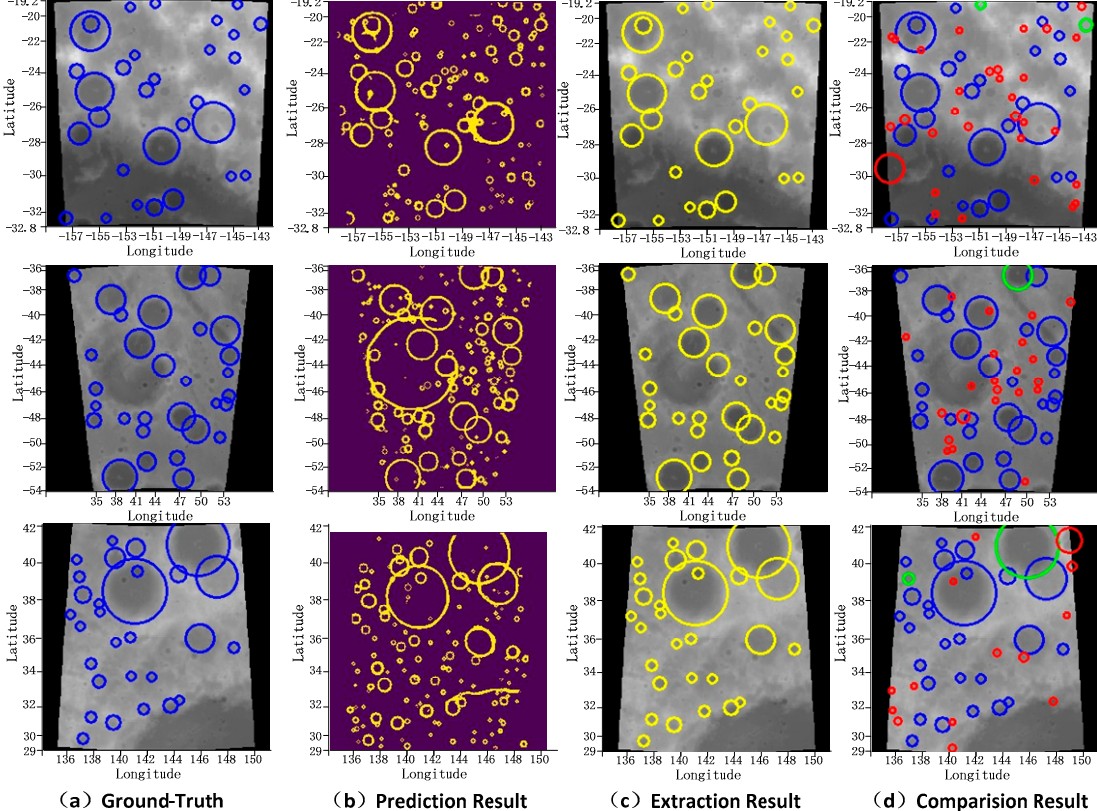

**Figure 3.** Crater prediction and extraction result by ERU-Net. (**a**) The ground-truth label of lunar images. Blue circles denote the ground-truth craters. (**b**) CNN (convolution neural network) prediction results. (**c**) Crater extraction results from prediction results. (**d**) The recognition result of our method is compared with the ground-truth. Blue circles denote the correctly recognized craters, red circles denote unrecognized craters, and green circles denote new craters that are predicted by our network.

The details of ERU-Net are as follows. Each residual unit contains three $3 \times 3$ convolution layers. We use the zero-padding to ensure that the size of output images is the same as that of the input images. The first convolution layer is used to adjust the number of channels. The second and third convolution layers yield a residual convolution block. In this block, we use the add operation to add the input feature map and output feature map obtained by performing the convolution two times. In the encoder part, we use the maximum pooling (max-pooling) operation to down-sample the output feature map from the residual unit. The pooling size is $2 \times 2$, and the size of the feature map is reduced to half of its original size. In the decoder part, we use $3 \times 3$ deconvolution with 2 strides as up-sampling layers to enlarge the size of the feature map. Then, the feature map is obtained by concatenating the deconvolution and its corresponding feature map of the same size in the encoder part. Note that the filling strategy in U-Net [18] convolution layers is not filling (valid filling), which may lead to the inconsistent image sizes. The input size of U-Net is $572 \times 572$, and the output size is $388 \times 388$. It is necessary to crop the images in the encoder part and then concatenate their associated feature maps in the decoder part. Since the filling strategy used in the ERU-Net convolution is zero-padding, the size of the feature map after deconvolution in the decoder part is the same as that of the corresponding feature map in the encoder part. They can be directly concatenated without cropping. The network depth of U-Net is 5, and the number of initial filters is 64. The network depth of ERU-Net for lunar crater recognition is 4, and the number of initial filters is 112. Inspired by a wide residual network (WRN) [30], ERU-Net uses dropout operation [31] after each max-pooling and concatenation operation. Using dropout in CNN can avoid the model falling into the trap of over-fitting. When the last residual convolution operation is completed, the number of channels in the output feature map is adjusted to 1 by a $1 \times 1$ convolutional layer, and then the output is activated by the sigmoid function to obtain the final output image.

Unlike deep residual U-Net [29], our ERU-Net uses $3 \times 3$ convolution to adjust the number of network channels before the residual unit, which can simplify the identity mapping procedure in the residual convolution [21]. The input and output sizes of each residual convolution are identical in ERU-Net. Therefore, they can be directly added without $1 \times 1$ convolution to adjust channel number and, consequently, can speed up the network training processing. In addition, deep residual U-Net uses $3 \times 3$ convolution with 2 strides to shrink the size of the feature map and exploits the up-sampling operation to enlarge the feature map size. In the proposed ERU-Net, the max-pooling operation is used for down-sampling, and $3 \times 3$ deconvolution with 2 strides is used for up-sampling.

### 2.3. Loss Computation

The loss function is used to estimate the inconsistency between predicted results and the ground-truth in machine learning [32]. The smaller the loss value, the better the robustness of the model.

The essence of the crater network prediction is to determine whether each pixel is on the crater edge. It is essentially a binary classification problem. Here, the loss function used in ERU-Net training process is binary cross-entropy (BCE) loss [33,34]:

$$loss = p_i - p_i t_i + log(1 + exp(-p_i)) \tag{1}$$

where $p_i$ is the label of a pixel $i$ in ERU-Net predicted result, and $t_i$ is the label of this pixel in the ground-truth. The loss of an image is the sum of the loss of all pixels. If the difference between the predicted image and the labeled image is large, the loss will be greater.

### 2.4. Crater Extraction

The image predicted by the network cannot directly provide crater information, such as the location information of craters. It is necessary to find out the location and size of the potential impact craters in the prediction result. Most impact craters are circular. We use the match template algorithm in scikit-image [35] that is an image processing package to match the crater edge. The match template

algorithm is a simple and direct algorithm. Firstly, we draw a circle with radius *Ra* in the blank image as a template, and the prediction result of the network is regarded as the target image. Then, the match template algorithm is used to get matching results. In the match template algorithm, we need a matching threshold to eliminate unreliable matching results. This threshold is 0.5, and the radius *Ra* ranges from 5 to 40 pixels in this work. Due to the existence of many overlapping craters, we find that the detection of rings in segmentation results using Hough transform is not so effective because it takes more time than a match template algorithm when setting a small center distance value. It is believed that this matching method is more accurate than other methods, i.e., Hough transform, and Canny edge detection [36].

In [37], researchers generated templates from LOLA (lunar orbiter laser altimeter) track data. The templates are crater images. Our match template algorithm is not the same as that in [37]. Our template is obtained by drawing circles rather than finding craters in segmentation results. [37] matched the impact crater directly from the original LOLA track image. We may get multiple impact craters at one location. So, we need to choose the most suitable crater for this location and use NMS to filter out other craters.

After extracting the impact crater, we need to determine whether the crater is correctly recognized. Given a lunar image $I$, $(x_i, y_i)$ is the position of a crater $c_i$ extracted from $I$ by our network, where $x_i$ is the latitude of the crater, and $y_i$ is the longitude of the crater. Let $r_i$ be the radius of the crater $c_i$. $(\hat{x}_j, \hat{y}_j)$ is the position of the ground-truth crater corresponding to $c_i$, where $\hat{x}_j$ is the latitude of this ground-truth crater, and $\hat{y}_j$ is the longitude of it. Its radius is denoted as $\hat{r}_j$. If Equations (2) and (3) are satisfied, the recognized crater is a correct crater. Otherwise, it is considered as a false crater. Denote $D_{x,y}$ as the longitude and latitude error threshold, and $D_r$ as the radius error threshold. In the experiments, we set $D_{x,y} = 2.0$ and $D_r = 1.0$.

$$\left( \left( x_i - \hat{x}_j \right)^2 + \left( y_i - \hat{y}_j \right)^2 \right) / min\left( r_i, \hat{r}_j \right)^2 < D_{x,y} \tag{2}$$

$$abs\left( r_i - \hat{r}_j \right) / min\left( r_i, \hat{r}_j \right) < D_r \tag{3}$$

### 2.5. Evaluation Method

The evaluation method used in the general semantic segmentation task is IOU (intersection over union) [38] of the network. The IOU score is the standard performance measure for the semantic segmentation. However, IOU is not suitable for the crater recognition task because we need to extract craters from prediction results. Therefore, we use the evaluation method commonly used in machine learning as follows.

Let $P$ be the precision and $R$ be the recall [39], which are commonly used to evaluate the performance of the classification model in machine learning. The precision and recall are, respectively, computed in Equations (4) and (5),

$$P = T_p / \left( T_p + F_p \right) \tag{4}$$

$$R = T_p / \left( T_p + F_n \right) \tag{5}$$

The craters satisfying Equations (2) and (3) are marked as correctly recognized craters (blue circle in Figure 3d), and the total number of these craters is denoted as $T_p$. Except for the above-recognized craters, craters in the prediction result that do not match ground-truth craters are marked as newly discovered craters [36] (green circle in Figure 3d), and their total number is denoted as $F_p$. The number of the unrecognized craters in the ground-truth crater set is denoted as $F_n$ (red circle in Figure 3d). $T_p + F_n$ is the number of ground-truth craters. $T_p + F_p$ is the number of predicted craters.

Generally speaking, when the precision is high, the recall is often low, and vice versa. To balance the influence of precision and recall, F-score $(F_\beta)$ [39] is used to measure the classification performance

of the model. When the recall is more important, set $\beta > 1$. When the precision is more important, set $\beta < 1$. When the recall and precision are equally important, set $\beta = 1$. $F_\beta$ is defined as follows,

$$F_\beta = \left(1 + \beta^2\right) \times P \times R / \left(\beta^2 \times P + R\right) \tag{6}$$

Note that the lunar crater is not completely annotated. That is, many truly existing craters are not marked in the ground-truth. These craters are treated as false-negative. Some examples of these craters are shown in Figure 1, in which they are denoted as the red circles in the first row. In [40], the airbus ship detection competition was evaluated using the $F_2$-score. The ground-truth data are incomplete in this competition that pays more attention to the recall measure. In our crater recognition, the crater information is provided by Head [25] and Povilaitis [26], and the label of the training dataset is incomplete too. Like [40], we also pay more attention to the recall measure. In this experiment, we set $\beta = 2$ and choose $F_2$-score as the main measure of network performance.

$$F_2 = 5 \times P \times R / (4 \times P + R) \tag{7}$$

In the previous section, the newly discovered crater is found by the network prediction. The false-positive rate of crater recognition in this paper is also called the discovery rate ($DR$) [36]. We use two discovery rates here. The first discovery rate ($DR_1$) is the ratio of the newly discovered craters to all recognized craters. The second discovery rate ($DR_2$) is the ratio of the number of newly discovered craters to that of all impact craters (recognized and unrecognized craters). They are defined as follows:

$$DR_1 = F_p / \left(F_p + T_p\right) \tag{8}$$

$$DR_2 = F_p / \left(F_p + T_p + F_n\right) \tag{9}$$

Besides the above important measurements in the lunar crater recognition, the accuracy of the position and size of the recognized craters can also evaluate the performance of the model. We calculate the latitude error ($Error_{la}$), longitude error ($Error_{lo}$), and radius error ($Error_r$) of recognized craters by the following formula.

$$Error_{lo} = abs\left(lo_p - lo_t\right) \times 2 / \left(r_p + r_t\right) \tag{10}$$

$$Error_{la} = abs\left(la_p - la_t\right) \times 2 / \left(r_p + r_t\right) \tag{11}$$

$$Error_r = abs\left(r_p - r_t\right) \times 2 / \left(r_p + r_t\right) \tag{12}$$

where $lo_p$ is the longitude value of the predicted crater, and $lo_t$ is the longitude value of the corresponding true crater. $la_p$ is the latitude value of the predicted crater, and $la_t$ is the latitude value of the corresponding true crater. $r_p$ is the radius value of the predicted crater, and $r_t$ is the radius value of the corresponding true crater.

## 3. Results

The experimental data in this paper are the lunar digital elevation model images from NASA [41]. Our method is compared with the CNN proposed by Aris et al. [36], deep residual U-Net 29, and D-LinkNet [42] by adjusting the number of training images, modifying the number of starting filters, and deepening the network structure. We use Keras [33] as the deep learning framework. All models are trained on one NVIDIA GTX1080 GPU with 8 GB onboard memory.

### 3.1. Training

We use the deep learning framework Keras [33] based on Python to construct the network model and use the Adam optimizer [43] to optimize the model. The learning rate is 0.0001, and the dropout rate is 0.1. For each batch, three images are trained during the training procedure. The training epoch is 5.

## 3.2. Experimental Results

### 3.2.1. Performance Comparison of Networks

In this section, we have conducted the experiments with different networks (ERU-Net, deep residual U-Net [29], the network proposed by Aris [36], D-LinkNet [42], ERU-Net-2, i.e., the residual unit of ERU-Net contains two Res-Blocks). The number of training images is 30,000. We randomly run each algorithm five times. The number of validation images is 3000. The number of test images is 3000. Crater recognition results of different networks are shown in Figure 4.

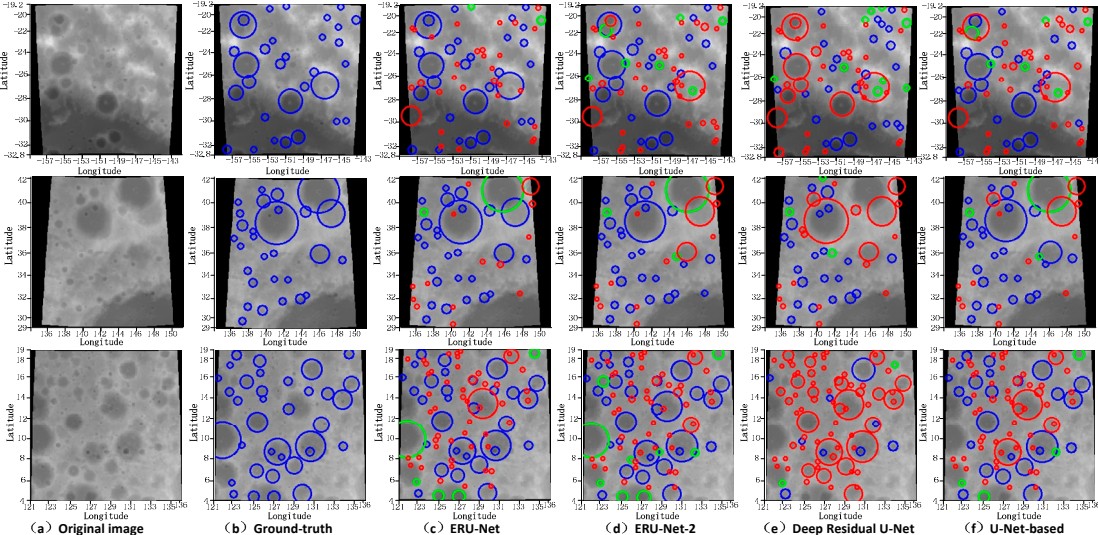

**Figure 4.** Crater recognition results of different networks. (**a**) The original lunar DEM (digital elevation model) data. (**b**) Ground-truth images. Blue circles denote ground-truth craters. (**c**) The recognition results of our proposed ERU-Net. (**d**) The recognition results of ERU-Net with two Res-Blocks. (**e**) The recognition results of deep residual U-Net [29]. (**f**) The recognition results of the network designed by [36]. In (**c**–**f**), red circles denote unrecognized craters, blue circles denote correctly recognized craters, and green circles denote new craters predicted by the network.

In order to demonstrate the performance of our algorithm, we evaluate the trained model by using the evaluation method mentioned in Section 2.4 on 3000 test images. The recognition result of each algorithm is shown in Table 1. It reports the mean of recall (Equation (4)), precision (Equation (5)), F2-Score (Equation (7)), discovery rates (DR1 in Equation (8), DR2 in Equation (9)), the latitude error denote as Lo-err and longitude error denote as La-err, and the radius error Ra-err (Equations (10)–(12)) obtained by the networks. For the first five measures, higher measure values indicate better recognition results, and for the last three measures, lower values indicate better recognition performance in Table 1.

**Table 1.** The recognition results of different network models on 30,000 training images (bold in the table means the best results on this evaluation index).

| Measures | ERU-Net | ERU-Net-2 | DRU-Net | Aris-CNN | D-LinkNet |
|----------|---------|-----------|---------|----------|-----------|
| Recall | **81.2%** | 80.2% | 76.7% | 76.1% | 68.3% |
| Precision | 75.4% | 77.5% | 77.9% | **83.2%** | 77.2% |
| F2-Score | **78.5%** | 78.4% | 76.9% | 76.5% | 67.7% |
| DR1 | 18.3% | 17.0% | **22.1%** | 13.3% | 17.3% |
| DR2 | **21.5%** | 19.3% | 18.3% | 13.7% | 17.1% |
| Lo-err | 9.9% | 9.6% | **7.6%** | 9.6% | 10.1% |
| La-err | 10.0% | 9.1% | **7.0%** | 9.2% | 10.0% |
| Ra-err | 7.8% | 7.7% | **4.8%** | 7.2% | 7.3% |

The number of initial filters in algorithms in Table 1 is 112. It can be seen from Table 1 that ERU-Net has a higher recall and discovery rate than the network structure proposed by Aris [36]. The recall of our method is about 5% higher than that of Aris' method. ERU-Net-2 has more parameters than ERU-Net because ERU-Net-2 has two Res-Blocks in the residual unit. The number of parameters of deep residual U-Net is almost equal to that of ERU-Net. We can see in Table 1, only DR1 of deep residual U-Net is higher than that of ERU-Net, and other measures of deep residual U-Net are lower than those of ERU-Net. For the Aris' Net, except for the precision measure, our method outperforms this network on the key measures in the lunar carter recognition. We can observe that among all compared approaches, our approach achieves the best recognition performance as a whole. As shown in Figure 5, we draw the P-R curves of ERU-Net, ERU-Net-2, and Aris' method. Match template thresholds are ranged from 0.2 to 0.8. When the threshold is increased, the recall of the model is increased, and the precision is decreased. It can be found that ERU-Net-2's performance is close to that of Aris'. ERU-Net's performance is better than others.

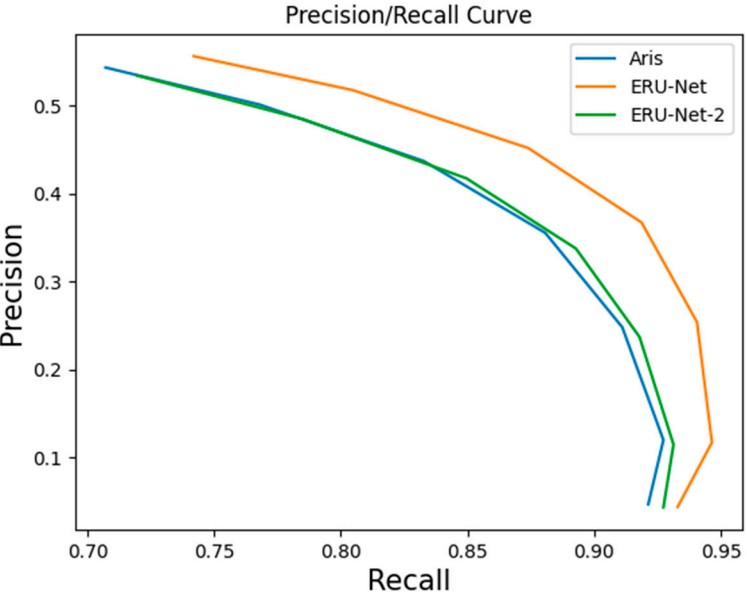

**Figure 5.** P-R curve of Aris' method and ERU-NET.

### 3.2.2. Fewer Train Data

In order to verify the learning ability of our proposed model, we change the number of training images to retrain the network model. The number of training images is 5000, and our model and Aris' network are run randomly for five times. The results are shown in Table 2. For each algorithm, this table also reports the mean of recall, precision, F2-Score, discovery rate of the network model, the latitude and longitude error, as well as the radius error.

**Table 2.** Recognition results of different network models on 5000 training images (bold in the table means the best results on this evaluation index).

| Measures | ERU-Net | ERU-Net-2 | Aris-CNN |
|----------|---------|-----------|----------|
| Recall | **74.5%** | 71.5% | 71.2% |
| Precision | 81.0% | **84.3%** | 80.2% |
| F2-Score | **74.7%** | 73.0% | 71.8% |
| DR1 | 14.7% | 12.4% | **15.2%** |
| DR2 | 15.2% | 11.9% | **15.3%** |
| Lo-diff | 10.6% | **10.0%** | 10.2% |
| La-diff | 9.6% | **9.1%** | 10.0% |
| Ra-diff | 7.6% | **7.5%** | 7.6% |

It can be concluded from Table 2 that the classification precision and recall of ERU-Net on 5000 images are higher than that of network proposed by Ari S et al. The highest precision is obtained by using ERU-Net-2, which is about 4% higher than the Aris' network. The recall of ERU-Net is 3.3% higher than that of the Aris' net. The ERU-Net-2 achieves the highest F2-Score in the three models. It can be found from Tables 1 and 2 that the recognition results obtained by using 30,000 training images are better than those obtained by using 5000 images. Moreover, we can observe that our proposed method is suitable for applying to a small-scale dataset.

### 3.2.3. Deepening Network Structure

In order to investigate whether adding residual blocks can effectively prevent performance degradation, we deepen the ERU-Net, deep residual U-Net [29], and Aris' Net [36]. The number of residual units in the encoder and decoder parts is three in ERU-Net and deep residual U-Net. Besides, the number of convolution units in the encoder and decoder parts is three in Aris' Net. We deepen ERU-Net and deep residual U-Net by adding a residual unit in the encoder and decoder parts, respectively. Similarly, we deepen Aris' Net [36] in the same manner. Before the deepening network structure, the depth of ERU-Net, deep residual U-Net, and Aris' Net is four. After deepening network structure, the depth of those networks is five. Considering the size of the GPU memory and the time costs, we set the number of initial filters to 56 in this experiment. The number of lunar images for training is 30,000. Besides, each network model is trained for five times.

Table 3 reports the recognition result after deepening the network structure. It can be seen from Table 3 that the recall of the ERU-Net increases by 1.8%, and F2-Score increases by 1.5% after deepening the network. In the deeper structure, the recall of deep residual U-Net increases by 1.2%, and the precision increases by 1.7%, but the F2-Score decreases by 1.4%. Note that the recall of the network in [36] decreases by 1.5%, the precision increases by nearly 1.7%, and the F2-Score decreases by 0.9% after deepening the network. Therefore, we can conclude that compared with deep residual U-Net and Aris' Net, ERU-Net achieves the best recognition performance after deepening the network structure. The reason is that ERU-Net uses residual convolution blocks instead of traditional convolution blocks, which are easily trained and more generalizable than traditional convolution blocks. The combination of Res-Block and batch normalization can perfectly solve the problem of gradient diffusion and sufficiently use classification information within images. Residual structure [21] can be used to solve the performance degradation problem of a deep convolution neural network under extreme depth conditions. The structure of the residual network is simple and can improve recognition performance. ERU-Net has a better recognition result if the network structure is deepened. Although deep residual U-Net uses the residual convolution, it achieves lower F2-Score and precision in this experiment, compared with our network.

**Table 3.** The recognition results of models after deepening the network (bold in the table means the best results on this evaluation index).

| Measures | ERU-Net-56 | ERU-Net-56-Deeper | DRU-Net-56 | DRU-Net 56-Deeper | Aris-CNN-56 | Aris-CNN-56-Deeper |
|---|---|---|---|---|---|---|
| Recall | 77.4% | **79.1%** | 76.7% | 77.8% | 76.6% | 75.2% |
| Precision | 81.3% | 81.7% | 69.9% | 60.4% | 81.7% | **83.2%** |
| F2-Score | 78.1 | **79.6%** | 75.2% | 73.6% | 77.6% | 76.6% |
| DR1 | 18.7% | 18.3% | 30.1% | **39.6%** | 18.3% | 16.8% |
| DR2 | 15.4% | 15.4% | 25.1% | **34.5%** | 14.9% | 13.5% |
| Lo-err | 7.4% | **7.3%** | 7.3% | 7.3% | 7.4% | 7.3% |
| La-err | 6.8% | 6.9% | 6.9% | 6.8% | **6.8%** | 6.9% |
| Ra-err | 4.9% | 5.0% | **4.8%** | 4.9% | 4.9% | 4.9% |

The number of network parameters of ERU-Net with 112 initial filters (denoted as ERU-Net-112) is 23.74 million, and the recognition results are shown in Table 1. In Table 3, our network has 5.9 million

parameters when it uses 56 initial filters (denoted as ERU-Net-56). The recall of ERU-Net-112 is 3.8% higher, and the F2-Score is 0.5% higher than ERU-Net-56. Hence, we can conclude that more network initial filters will lead to more network parameters and better recognition results.

### 3.2.4. Crater Distribution

The DEM images used in experiment 3.2.1–3.2.3 are randomly selected examples and are not necessarily completely representative of the network recognition algorithm's ability to recognize craters in areas with different crater densities. In this section, we discuss the recognized result on different density degrees. We divide the density of distribution into three levels, i.e., high, medium, and low. The number of craters, which are larger than 5 pixels in diameter, in a $256 \times 256$ image less than 20 is low. The number between 20 and 100 is medium, and the number of more than 100 is high. The experiment results are shown in Table 4. We can find when the number of craters increases, the recall decreases, and the precision increases. In Table 4, our ERU-Net achieves higher recall and precision than U-Net in high and medium levels. At a low level, the recall of ERU-Net is higher than U-Net, and the precision is 3.5% lower than U-Net.

In this experiment, we prove that our method also yields great recognition results in dense areas than others.

**Table 4.** The recognition results in different distributions (bold in the table means the best results on this evaluation index).

| Measures | ERU-Net | | | U-Net | | |
|----------|---------|-----------|----------|--------|-----------|----------|
|          | Recall  | Precision | F2-Score | Recall | Precision | F2-Score |
| High     | **66.7%** | **94.6%** | 70.8% | 62.4% | 93.8% | 66.8% |
| Medium   | **82.1%** | **83.1%** | 82.2% | 79.8% | 79.8% | 79.8% |
| Low      | **90.1%** | 56.4% | 80.4% | 85.2% | 59.9% | 78.5% |

### 3.2.5. Best Model of Our ERU-Net

In order to get the best model of our ERU-Net, we have spent five GPU Days to train ERU-Net with 112 initial filters on 50,000 training images. The depth of ERU-Net is four. For each batch, three images are trained. After an epoch is finished, we evaluate the trained model on 3000 test images. The evaluation results are shown in Tables 5 and 6. They show that when the running times of training increase, the precision tends to constantly increase. But, as shown in Table 5, we have found that the recall decreases between 25 and 30. Therefore, the best model appears between 25 and 30. Table 6 shows that the recall and F2-Score reach the maximum at the 28th training-time. The recall of our model based on ERU-Net with 112 initial filters achieves 83.59%, and the precision is 84.80% on 3000 test images.

**Table 5.** The recognition result between 15 and 30 training time (bold in the table means the best results on this evaluation index).

| Epochs<br>Measure | 15 | 20 | 25 | 30 |
|---------|---------|---------|---------|-----------|
| Loss | 0.06021 | 0.05916 | 0.05789 | **0.05667** |
| Recall | 82.2% | **83.2%** | 83.1% | 82.9% |
| Precision | 83.1% | 83.6% | 84.8% | **85.3%** |
| F2-Score | 81.5% | 82.5% | 82.6% | **82.6%** |
| DR1 | **13.3%** | 12.9% | 12.1% | 11.7% |
| DR2 | **14.6%** | 14.2% | 13.2% | 12.7% |
| Lo-err | 9.5% | 9.3% | 9.1% | **8.9%** |
| La-err | 9.5% | 9.2% | **8.7%** | 8.7% |
| Ra-err | 7.5% | 7.4% | 7.5% | **7.0%** |

**Table 6.** The recognition result between 26 and 29 training time (bold in the table means the best results on this evaluation index).

| Epochs<br>Measure | 27 | 28 | 29 | |
|---|---|---|---|---|
| Loss | 0.057456 | 0.056912 | **0.056792** | 0.056800 |
| Recall | **83.7%** | 83.5% | 83.6% | 83.3% |
| Precision | 84.0% | 84.8% | 84.8% | **85.3%** |
| F2-Score | 82.9% | 83.0% | **83.1%** | 82.9% |
| DR1 | **12.6%** | 12.1% | 12.1% | 11.7% |
| DR2 | **14.1%** | 13.2% | 13.2% | 12.7% |
| Lo-err | 8.9% | 9.1% | 9.0% | **8.8%** |
| La-err | 9.1% | 8.9% | 8.6% | **8.3%** |
| Ra-err | 7.4% | **7.1%** | 7.2% | 7.3% |

## 4. Discussion

The proposed method achieves high recall and precision on Moon crater recognition. It is a practical algorithm for recognizing impact crater. In the previous section, we have reviewed the existing methods and compared them with our method. In this section, we discuss our method from three perspectives. First of all, we discuss the method of image semantic segmentation. Secondly, the crater extraction method is discussed. Finally, we discuss our future work and the improvement direction of the impact crater recognition.

### 4.1. Discussion: Experiment Result

As shown in Section 3.2., the algorithm of impact crater recognition proposed by us achieves good recognition results. The recall rate of our algorithm is improved by 5% compared with Aris' method [36]. We propose an image segmentation network called effective residual U-Net. The segment ability of this network on the crater recognition task is better than deep residual U-Net [29], the network proposed by Aris [36], and D-LinkNet [42]. ERU-Net has a strong learning ability and is suitable for applying on the small-scale dataset. After deepening the network structure, our method can achieve a 2% recall improvement. Deep residual U-Net only shows a 1.2% improvement. U-Net shows a 1.5 reduction. For the impact craters with different density distribution, our method can achieve good recognition results. The reason why our method is better than the others is that our method has better segmentation results. The network we have designed has better learning ability than others.

### 4.2. Discussion: Image Segment

This method attempts to use an image segmentation method to extract the edge of impact crater from the DEM of the Moon and then achieve the purpose of recognizing impact crater. We use residual convolution block, dropout, batch normalization, and other skills to improve U-Net. Our results suggest that using residual convolution can improve the segmentation performance of the network. In the case of a small amount of train data, the learning ability to use residual convolution is strong. When increasing the number of network layers, the use of residual convolution can get better segmentation results, whereas the segmentation result of the network without residual convolution is decreased after deepening the network. Nevertheless, it doesn't mean that the more residual convolution we use, the better is the segmentation result. For example, the result of ERU-NET-2 is not as good as using ERU-NET when the initial filter number is 56.

In general, the residual structure solves the problem of gradient vanishing and gradient exploding in a deep convolution network since it uses skip connection and identity mapping to enhance the learning ability of the network.

*4.3. Discussion: Crater Extraction*

In this work, we use the template matching algorithm to extract the impact crater from the segmentation results, which is better than other algorithms (Canny edge, Hough transform). It can extract the small crater in the big crater. We can use smaller thresholds to identify stacked impact craters. However, the Canny edge and Hough transform are difficult to deal with these situations. They are suitable for dealing with independent and separated impact craters.

## 5. Conclusions

In this paper, an effective crater recognition algorithm is proposed to recognize craters from the lunar digital elevation model. The research will help lunar researchers identify impact craters more quickly and map the Moon's topography. We use image segmentation and template matching to solve the problem of overlapping crater recognition and propose a new deep learning method, i.e., ERU-Net, which is based on U-Net and the residual convolution to perform lunar crater recognition. This work demonstrates that using the proposed ERU-Net to recognize craters from the lunar digital elevation map is effective and feasible. Compared with other deep learning methods applied in the lunar crater recognition, our ERU-Net outperforms these compared methods. Moreover, our network can achieve better recognition results when the network structure is deepened. In addition, ERU-Net also can be used in other image segmentation tasks.

In future work, we will continue to study in three areas. First, we will improve the shot-connections of ERU-Net with [44–47] or apply some methods of instance segmentation [48,49] to recognize crater directly. We will apply our method combining with some traditional classification methods, such as sparse representation algorithm [50], to the impact crater recognition on other celestial bodies. Second, multi-information fusion is also a problem worthy of attention. We will try to use our method to experiment on ordinary photographic images in the future and find a method to combine DEM features and ordinary photographic image features to recognize impact crater. Third, we will seek or design better impact crater extraction methods to deal with the more complex stack craters and improve extraction results.

**Author Contributions:** Conceptualization, S.W. and H.Z.; methodology, S.W.; software, S.W.; validation C.W.; formal analysis, S.W.; investigation, Z.F.; resources, S.W.; data curation, Z.L.; writing—original draft preparation, S.W.; writing—review and editing, Z.F.; visualization, S.W.; supervision, Z.L.; project administration, Z.F.; funding acquisition, Z.F. All authors have read and agreed to the published version of the manuscript.

**Funding:** This research was funded by the Natural Science Foundation of China, grant number 61991401, 61673097, 61490704, 61702117 and Jiangxi Provincial Natural Science Foundation of China, grant number 20192ACBL20010.

**Acknowledgments:** Our sincere thanks to the Key Laboratory of Advanced Control and Optimization of Jiangxi Province support for this paper. The authors also gratefully acknowledge Charles Zhu for the dataset.

**Conflicts of Interest:** The authors declare no conflict of interest.

## Appendix A

*Moon LRO LOLA DEM 118m v1*

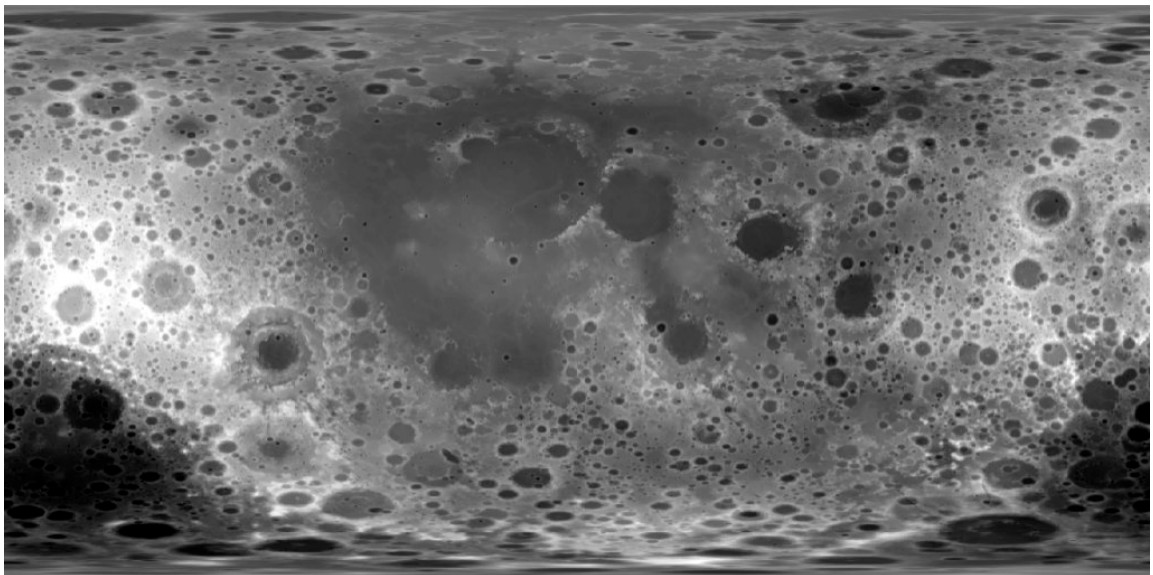

**Figure A1.** The lunar DEM is based on data from the lunar orbiter laser altimeter [51]. More than 6.5 billion measurements gathered between July 2009 and July 2013 [52] are used to build this DEM dataset. The elevations in this DEM are computed by subtracting the lunar reference radius of 1737.4 km from the surface radius measurements [53,54].

## Appendix B

*Crater Statistics*

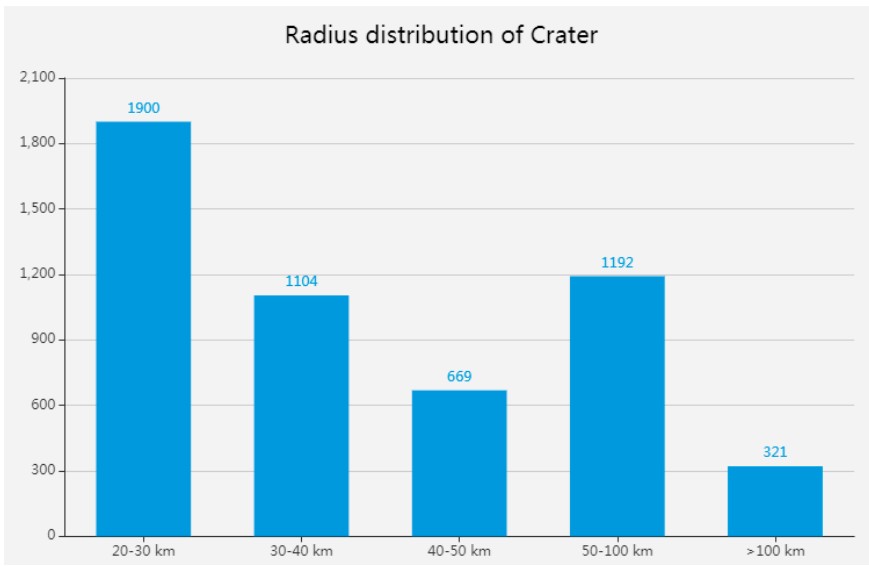

**Figure A2.** Distribution of large lunar craters (radius > 20 km) made by Head [25]. The total number of impact craters is 5186.

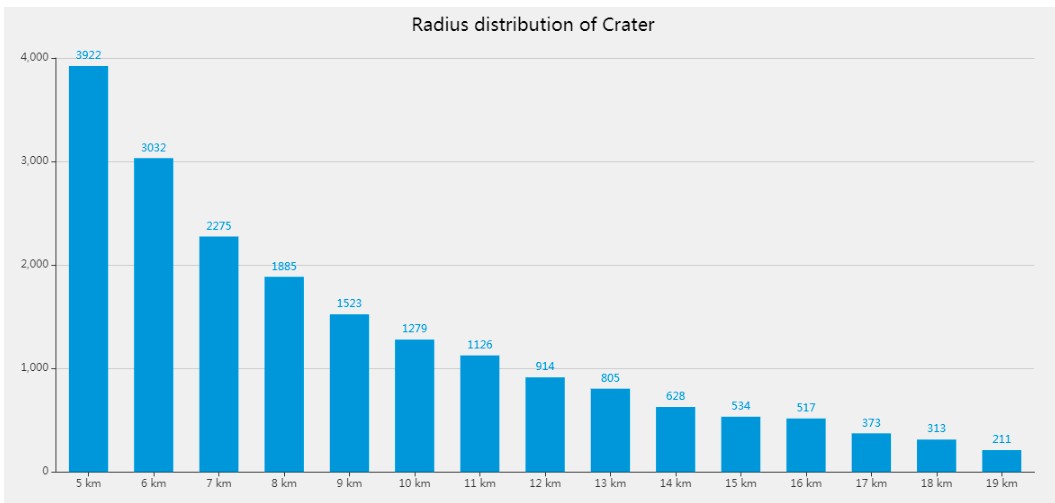

**Figure A3.** Distribution of small lunar craters (radius < 20 km) made by Povilaitis [26]. The total number of impact craters is 19,337.

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
