# Peer review of "An Effective Lunar Crater Recognition Algorithm Based on Convolutional Neural Network"

_remotesensing, doi:10.3390/rs12172694_

Round 1
Reviewer 1 Report
Review of “An effective lunar crater recognition algorithm based on convolutional neural network”
Authors: Song Wang, Zizhu Fan, Hong Zhang, and Chao Wei
Reviewer: Brad Thomson (Univ. of Tennessee)
Summary of contribution
This manuscript describes the results of the application of a new computer automated method using convolulational neural network to recognize impact craters on the surface of the Moon. This method is applied to lunar topographic data from the Lunar Reconnaissance Orbier mission, and two crater databases of manually-measured craters are used as ground-truth data. Different crater recognition algorithms are tested and compared, and the results note that the presented methodology is an improvement over other previously published methods.
Summary of review
As a planetary geologist who uses the size-frequency distributiuon of craters on bodies like the Moon to help infer the nature and timing of resurfacing processes, I am excited at the prospect of having automated method to aid in this technique. Despite my enthusiasm for the overall goal of this work, and I am less than fully enthusiastic about the results presented in this manuscript because I feel that the work raises more questions than it answers. In particular, many detailed aspects of the methodology are unclear. I have attempted to flag all areas that could use some substantial improvement as “major” in the comments below.
Second, while I appreciate the authors making their code available, it would also be helpful to make their training data and the results given in the figures of this manuscript available as well. The lack of latitude and longitude labels on the figures make these difficult to impossible to reproduce.
Third and finally, I would like to see more analysis of the model output. I am still a little unclear if the green craters given in Figures 2 to 4 are actual craters that were not included in the ground-truth data, or if they are false positives. Are there any systematic trends in the false negatives or false positives? Do they occur equally across all diameters, or it is mostly smaller craters (i.e., those that are coarser in the data). How well are overlapping craters disentangled? A better description of these points would indicate where on the Moon and under what circumstances this method could be confidently applies, and conversely where the results should be treated with caution.
Numbered points to consider (the label “MAJOR” indicates those points that must be addressed prior to publication, otherwise they are minor suggestions).
- Line 34. The phrase “lack of geological and atmospheric activity on the Moon” needs some clarification. While the Moon lacks common terrestrial geological processes due to wind and water, impact cratering is a geologic process. Volcanism is also abundant on the Moon, and is also a geologic process.
- Non-existent reference. Line 37. Reference #1. My original comment was that this reference seemed too specific. As per the title “The morphological classification and distribution characteristics of craters in the LQ-4 Area,” this title appears to be about a specific region, not about the importance of craters on the Moon as a whole. But there are additional problems with this reference: it does not appear to exist. The journal “Earth science Frontiers” was a Springer journal that ceased publication in 2010, yet this citation is from 2012. There is a another Springer title based in China called “Frontiers of Earth Science,” but there were only 4 issues published in 2012 (the citation issue number is 6), and no similarly titled articles could be found.
- Lines 39–40. Here it is stated that “second craters tend to cluster in a ring around the main crater.” It is true that secondaries cluster, but they are concentrated into discrete rays, not an annual ring.
- Line 41–42. I disagree with the assertion that secondary craters are “usually so small that [they] are hardly recognized.” The size of secondary craters is proportional to the size of the parent crater. There are plenty of large craters on the Moon whose secondaries are clearly discernible (i.e., more than 10 pixels across) in high spatial resolution optical images.
- Lines 48–49. Regarding the discussion of interpersonal variability in manual crater recognition, I suggest citing Robbins et al. (2014).
Robbins, S.J., Antonenko, I., Kirchoff, M.R., Chapman, C.R., Fassett, C.I., Herrick, R.R., Singer, K., Zanetti, M., Lehan, C., Huang, D. and Gay, P.L., 2014. The variability of crater identification among expert and community crater analysts. Icarus, 234, 109–131. - Lines 52–77. This paragraph recants a number of recent publications that make use of automated crater detection algorithms, references 2 through 8. My question is why aren’t these in chronologic order? The rationale for the ordering of these citations is not clear to me.
- Lines 111–114. Here, some of the results are described in the introduction section. This runs contrary to the standard breakdown of what material appears where. Typically, the introduction only lays out the rationale for the study, not the results themselves. Suggest omitting.
- Lines 190 and 191. Please insert “The” before “match.” The match template algorithm is described as “simple and force.” Perhaps “simple and straightforward” was intended? Please clarify.
- [MAJOR?] Lines 194–105. It is mentioned that the crater radii Ra considered in this work ranged from 5 to 40 pixels. But what are the physical dimensions of these craters? This also leads into my question about the data project used, because the pixel scale of a cylindrical projection (e.g., as depicted in Figure A1) is not constant in longitude.
- [MAJOR] Inappropriate spatial criterion. Lines 212–213. The longitude and latitude error threshold (Dx,y) is stated to be 2.0. Is this really two degrees? Because on the Moon, one degree of latitude is 30.3 km, so two degrees would be 60.6 km. This would seem to be an unacceptably large criterion. What is the size range of craters considered? Figure A2 suggests >20 km, and Figure A3 suggests >5 km. Even considering the larger limit, this overly liberal lat-lon threshold permits an error of 300%. In a real word case, having a coordinate offset more than a few percentage points off the true value would certainty constitute a false detection.
- Line 213. What are the units of the radius error threshold? Need to link to a spatial dimension in order to have geological meaning.
- [MAJOR] Lack of appropriate figure labels and descriptions, also shaded relief maps are required. Line 220, Figure 2. There are several issues with this figure. First, latitude and longitude labels are needed, as is a scale bar. Is the bottom area depicted Crisium Basin or Orientale Basin? Second, these figures would be much clearer if craters were mapped onto a shaded relief background rather than a grayscale image shaded by elevation. When looking at “raw” topography values, it is extremely difficult to discern subtle topographic signatures. This is why the standard practice is to prepare a shaded relief map. A basic standard for any journal figure is that it should be reproducible by another researcher. Without appropriate labels, these figures cannot be verified or reproduced. Finally, what projection(s) is/are used in these figures? It would appear that unlike Figure 1A, these are not cylindrical projections. The exact projection used should be specified so that it could be reproduced by others.
- [MAJOR] Ground-truth data are (a) described and (b) obviously incomplete. Line 220, Figure 2. A separate concern with this figure is what is the source of the “ground-truth” craters? Are they drawn from an existing crater catalog? Because the number of craters evident in the ground-truth panels look far too incomplete. If one examined a shaded relief map, it would be evident that there are additional, unmapped craters present by a factor of likely several.
- [MAJOR] Ground-truth completeness unclear; newly identified craters unverified. Line 228. Equation 4 contains the variable Fp, and Equation 5 contains the variable Fn. The former is described as “newly discovered craters” (line 231) and “unrecognized craters” (lines 232–233), respectively. Are these the same as false positives and false negatives, as the variables names would imply? Also, this point raises a question in regards to Fig. 2d. The green circles in Fig. 2d represent “new craters that are predicted by our network,” which are presumably the same as those designated Fp in Equation 4. Are they false positives, or are they newly detected, actual impact structures that weren’t included in the ground-truth data set? This is why I asked earlier if the ground-truth data set was complete, i.e., contained all readily identifiable craters down to some specified threshold diameter. As a separate point, one major thing that is lacking in this work is a verification step to determine the crater detections given in green in Fig 2d are actual craters or false positives.
I see some discussion on this point in the paragraph beginning on Line 244, but this information needs to be moved up earlier in methods. This description does not make sense to me. Why are actual impact structures that are not included in the ground-truth set considered false negatives? - Line 240, Figure 3. As with Fig. 2, more information is needed about latitude and longitude limits, the projection or projections used, scale bars, and finally shaded relief maps. The same is true for Figure 4 as well.
- [MAJOR] Out-of-order information. Lines 272–287. This section, entitled, “Experimental Data” seems to be in the wrong section. Here, the topographic information and crater databased used are described, but this is in section 3 “Results.” This material should be in the methods section.
- [MAJOR] Characteristics of input DEM insufficiently described. Line 275. Here it is stated that the pixel size of the DEM used for this study is 92,160 by 30,720 pixels. This is a bit confusing at it implies a 3:1 width-to-height ratio. What is the pixel scale? What is the extent of this data set: 0 to 360°E, 80°S to 80°N? Finally, what is the projection? Is a single projection used for the whole data set? In the high latitude panels in Figures 2 through Figure 4, different centers of longitude appear to be used for these non-cylindrical projections. How were these different projections handled in the processing, or were these just used to display the results after the fact?
- Lines 279–280. Crater database description insufficient. There is a brief mention of two crater databases by Head et al. [2010] and Povilaitis et al. [2018] (note wrong year in given in the citation list line 538). However, this terse description is not sufficient. What data were used to compile each crater database? To what diameter are they considered “complete”? It would seem that the authors intended to provide some of this information in Appendix A, but it is not referenced here in the text.
- Lines 290–292. What are the units for the learning rate, dropout rate, and training epoch?
- Lines 279–299. The text notes that the number of training images is 30,000, and that the number of test images (in this component) is 3,000. Perhaps this question reflects my limited understanding of this methodology, but how many craters are their per image? One, or several?
- Lines 309–310. The rows of Table 1 are described here. I suggest referencing the equations given in the methodology section as an aid to readers.
- Line 313, Table 1 title. In either the description of this table or a table footnote, it would be good to explain to the readers what the values given in bold mean. It is not necessarily obvious at first glance that the bold values for the first five rows under the title give the maximum value, while in the last three rows, values given in bold are minimum values.
- Line 313, Table 1. Perhaps this is common practice in the computer science field, but reporting all values in this table to four significant figures feels like overkill. Three significant digits would likely be sufficient as I am doubtful the authors would ascribe much significance to results that only differed in the hundreds decimal place.
- Line 318–319. Here it is mentioned that the training speed of one method is 50% faster than the other. Perhaps I missed it, but I don’t see any documentation of execution speeds, hardware used, etc.
- Lines 381–382. Suggest rephrasing this sentence. Original: “The above experimental images are obtained by random clip DEM image, which cannot reflect the network recognition ability in different crater density.” Suggested revision, example only: “The DEM images given in Figures 2–4 are randomly selected examples, and are not necessarily completely representative of the network recognition algorithm’s ability to recognize craters in areas with different crater densities.”
- Line 384. Here it is stated that “The number of craters in an image less than 20 [is categorized as] low.” This quantitative threshold needs to be reported as the cumulative number larger than a particular crater diameter (N>D) per unit area (e.g., 106 km2).
- Lines 402–403. Tables 5 and 6. It appears that some of the bolded entries given in these tables is not correct, or at least, the same convention used to assign bold values is not used for all rows. In the second row of Table 5, for example, the entry 83.07% is given in bold, but the value of the column to the left (column 20) has a higher value (83.17%), indicated this entry should be given in bold font, not the value in column 25. Similar bold errors occur in Table 6 row 2 (should be maximum value given in column 4) and row 7 (should be minimum value in column 5).
- Line 406. Please capitalized “Moon.”

Reviewer 2 Report
L 98 ‘Since the DEM image is hardly affected by the illumination and camera angle, our work will perform crater recognition on DEM’.
** I am confused with this sentence. Do you mean ‘optical image’ in the first statement?
Although English is not my mother tongue, the paper looks that it needs a significant improvement for its language e.g. ‘Finally, we compared the extracted craters with real craters.’
** Target matching looks it decreases the performance since you loose the biggest crater in the low-left side of the Fig.2. Maybe better to refine the methodology with also matching curves/half circles?
Line 244 you have mentioned ‘Note that the lunar crater is not completely annotated. That is, many truly existing craters are not marked in the ground-truth.’ Why?
** A comparison between the others is also needed for the whole processing time.
**Line 427 You say ‘in this work,we use the template matching algorithm to extract the impact crater from the segmentation results, which is better than other algorithms(Canny edge, Hough transform)’. But there is no result reported from these methods.
**In discussion, it is expected to mention the others work in terms of the outcomes and what makes your method is better than the others. Especially Aris’ method should be main work has to be mentioned.
Reviewer 3 Report
The submitted manuscript rigorously presents method of identifing craters on the Moon surface using its digital elevation model. I consider the research interesting to the readers of the journal, but the text of the manuscript has to be revised to more convincingly emphasize the novelty of the research undertaken. There are plenty of papers published on the topic. The presentation of the introduction, methods, discussion and conclusion requires revision. Minor grammer/spelling flaws are present.
I highly recommend to revise the introduction. The argument for the research undertaken is restricted to review of few publication but there are more which deal with the objective of the authors. I recommend to ellucidate the research gap from the view point of the following recent publications dealing with identification of craters using DEM data.
Lemelin, M., Daly, M.G., Deliège, A. Analysis of the Topographic Roughness of the Moon Using the Wavelet Leaders Method and the Lunar Digital Elevation Model From the Lunar Orbiter Laser Altimeter and SELENE Terrain Camera (2020) Journal of Geophysical Research: Planets, 125 (1), art. no. e2019JE006105, . DOI: 10.1029/2019JE006105
Włodarski, W., Papis, J., Szczuciński, W. Morphology of the Morasko crater field (western Poland): Influences of pre-impact topography, meteoroid impact processes, and post-impact alterations (2017) Geomorphology, 295, pp. 586-597. DOI: 10.1016/j.geomorph.2017.08.025
Vamshi, G.T., Martha, T.R., Vinod Kumar, K. An object-based classification method for automatic detection of lunar impact craters from topographic data (2016) Advances in Space Research, 57 (9), pp. 1978-1988.
DOI: 10.1016/j.asr.2016.01.022
Kang, Z., Luo, Z., Hu, T., Gamba, P.
Automatic Extraction and Identification of Lunar Impact Craters Based on Optical Data and DEMs Acquired by the Chang'E Satellites
(2015) IEEE Journal of Selected Topics in Applied Earth Observations and Remote Sensing, 8 (10), art. no. 7299593, pp. 4751-4761.
DOI: 10.1109/JSTARS.2015.2481407
DeLatte, D.M., Crites, S.T., Guttenberg, N., Yairi, T.Automated crater detection algorithms from a machine learning perspective in the convolutional neural network era(2019) Advances in Space Research, 64 (8), pp. 1615DOI: 10.1016/j.asr.2019.07.017 Zhou, Y., Zhao, H., Chen, M., Tu, J., Yan, L.Automatic detection of lunar craters based on DEM data with the terrain analysis method(2018) Planetary and Space Science, 160, pp. 1-11. DOI: 10.1016/j.pss.2018.03.003 Chen, M., Liu, D., Qian, K., Li, J., Lei, M., Zhou, Y.Lunar Crater Detection Based on Terrain Analysis and Mathematical Morphology Methods Using Digital Elevation Models(2018) IEEE Transactions on Geoscience and Remote Sensing, 56 (7), pp. 3681-3692. DOI: 10.1109/TGRS.2018.2806371
What is the innovation of the research done compared with the published studies? Please, emphasize the unique aspect of your study. Further in the discussion part, how do the results achieved by the authors compare with the performance of the published approaches?
The final paragraph of the Introduction should result into the argument of what is unique in the presented study and what is the aim and objective. The final paragraph is a mixture of brief overview of the methods and results.
I suggest to start the Mehtods with description of the data used, which is the final paragraph of the Introduction. Afterwards, you can describe the approach used.
The results should be discussed in the Discussion section also with regard to other publihsed methods which are referred to in the Introduction. Much is dicussed within the part of Results, which could be moved to the Discussion.
Conclusions are defined very briefly. The main point of the research undertaken need to be stated, the key benefits of the research. I would use the argument of using a DEM for crater recognition which is not directly stated in the conclusions. Also define the future work within the COnclusion, not in the Discussion.
Minor corrections refering to:
Captions of the figures should not preceed the reference in the text, e.g. Fig. 1.
I suggest to refrain from the term DEM image, I assume you worked with a digital elevation model dataset, not just an image of the DEM dataset. So, DEM tiles, DEM raster layers, DEM data etc. Would be more appropriate terms.
Sentences as in lines 115-117, 120-123 are redundant, the structure of the manuscript is clear from its titles.
Caption of Figure 2 starts wiht „Rater“, replace with „Crater“, I do not understand why large circular crater feature in the upper right corner of the Figure 2 bottom row is not used as the ground truth crater. Could you explain/refer to it in the text.
Reviewer 4 Report
Review for “An Effective Lunar Crater Recognition Algorithm Based on Convolutional Neural Network,” by Wang et al.
This paper describes a new automated method to detect lunar craters in a digital elevation model (DEM). The new method, called Effective Residual U-Net (ERU-Net), uses a convolutional neural network to recognize craters by detecting their edges and then matching templates to compute their positions. The training process of ERU-Net is relatively simple and easily optimized. Since the method focuses on the crater rim, it can recognize overlapping craters. The ERU-Net method achieves a high recall and precision of ~85% on a global lunar DEM. The recall of this method is higher than that of other deep learning methods.
Crater identification is an important technique in planetary science as it is used to age-date planetary surfaces. The paper is an interesting contribution to this field, and will be suitable for publication after addressing my comments and questions below.
Major Points
The authors state that their new method has a higher recall than other methods, but the difference does not seem all that large, only a few percent. Why go to all this trouble only to achieve a small improvement in recall? On a related note, it would be helpful if there was some way to judge the statistical significance of the percentages in the Tables, perhaps by repeating the experiments many times on different sets of images and computing the spread in percentages.
It is also not clear how practical this method would be in a “real-world” application, such as finding craters in a stereo-based DEM from the Lunar Reconnaissance Orbiter Narrow Angle Camera with a pixel scale of ~5 m/pixel. Would it be possible to generate a large-enough training set from such a DEM? (many examples are available for download on the LROC website).
I would like to see some discussion and a figure of the crater size-frequency distribution for the recovered craters compared to the distribution from independent previously-compiled catalogs. At what crater diameter does this distribution begin to flatten out or differ substantially from the actual distribution? This would be a useful metric to estimate the completeness limit of the method, which is an important quantity to know when age dating planetary surfaces. Also, how do the recall, precision, and other scores listed in the Tables vary with crater diameter? That would be very useful to know for understanding its performance.
Minor Points
Line 213: What are the units of Dxy and Dr?
Line 246: [1830] ?
Line 273: Give the DEM resolution in pixels/degree and meters/pixel at the equator.
Sec. 3.1: How did the authors account for projection effects which can make craters appear non-circular at high latitudes?
Line 379: Is it worth the extra time to add more network initial filters?
Table 4: Why aren’t the other methods listed in Table 4 as in 1 – 3?
Lines 423 – 425: Please clarify this sentence. What is meant by “gradient vanishing”, “gradient exploding”, “skip connection”?
General: Do the authors plan to run their method on the lunar DEM to produce a new global crater catalog for the lunar scientist community?
Reviewer 5 Report
The general impression is that the article is interesting.
The article proposes a neural network for identifying craters on the Moon. The description of the network is quite complete, the error function is generally accepted, the proposed network architecture is quite innovative. The article contains a link to the source code 8 months ago, but I could not run it, perhaps I did not have enough experience.
There are no complaints about the design of the article.
Remarks.
- I recommend to authors add links to articles on related topics in the introduction. For example on these: http://arxiv.org/abs/1912.01240, http://arxiv.org/abs/1904.07991. And compare them with the approach of the authors.
- The authors chose the digital elevation model as a source of information about the Moon. This source is more informative than conventional (2-dimensional) photographs of the Moon, but much more rare. In systems of circumlunar orientation, surface recognition will be carried out precisely from ordinary photographic images. I would like the authors to provide more detailed information about the possibility of their neural network working with photographic images of its surface.
Round 2
Reviewer 1 Report
Attached, please find my review of Manuscript ID: remotesensing-874977. The authors have done a thorough job of responding to my questions, and I thank them for their efforts. I have a few outstanding issues highlighted in blue in the attached document. The two biggest are as follows. (1) Mismatch between crater database size and training image size in pixels. The crater databases used have a minimum crater diameter of 5 km. Yet the review response notes that "the range of radius Ra from 5 to 40 is the best numerical range finally obtained by template matching." With 59 m/pixel data, this works out to 295 to 2360 m in radii or 0.59 to 4.720 km in diameter. These two spatial scales do not overlap, hence my confusion. (2) Please add lat/lon labels to Figures 2 and 3 so that these example locations are not a mystery. Many thanks for clarifying these last items.

Author Response
Dear Reviewer.
Thank you for your careful review and helpful comments concerning our manuscript entitled:“An Effective Lunar Crater Recognition Algorithm Based on Convolutional Neural Network”. Those comments are all valuable and very helpful for revising and improving our manuscript, as well as the important significance to guide our research. We have studied comments carefully and have made revision which we hope to meet with approval. The revisions are marked by using “Track Changes” in the new manuscript. The main revisions in the manuscript and the responses to the reviewer’s comments are as follows

Reviewer 2 Report
very minor comment: You have revised the numbers in the tables, but they have to be rounded to upper value, e.g.76.69% has to be 76.7 not 76.6
Author Response
Point 1: You have revised the numbers in the tables, but they have to be rounded to upper value, e.g.76.69% has to be 76.7 not 76.6
Response 1: Thank you for your helpful suggestion and careful reviews. We accept your suggestion. The numbers in revised tables are rounded to upper value. It can be found in Tables 1 to 6.